# Analysis of Flavonoid Metabolites in Citrus Peels (*Citrus reticulata* “Dahongpao”) Using UPLC-ESI-MS/MS

**DOI:** 10.3390/molecules24152680

**Published:** 2019-07-24

**Authors:** Fu Wang, Lin Chen, Hongping Chen, Shiwei Chen, Youping Liu

**Affiliations:** 1Department of Pharmacy, Chengdu University of TCM, Standardization education ministry key laboratory of traditional Chinese medicine, Chengdu 611137, China; 2Food & Drugs Authority of Nanchong, Nanchong 637000, China

**Keywords:** flavonoid metabolites, UPLC-ESI-MS/MS, *Citrus reticulata* “Dahongpao”, citrus peels, PCA, OPLS-DA

## Abstract

Flavonoids are a kind of essential substance for the human body because of their antioxidant properties and extremely high medicinal value. *Citrus reticulata* “Dahongpao” (DHP) is a special citrus variety that is rich in flavonoids, however little is known about its systematic flavonoids profile. In the present study, the presence of flavonoids in five important citrus varieties, including DHP, *Citrus grandis Tomentosa* (HZY), *Citrus ichangensis Swingle* (YCC), *Citrus sinensis (L.) Osbeck* (TC), and *Citrus reticulata* ‘Buzhihuo’ (BZH), was determined using a UPLC-ESI-MS/MS-based, widely targeted metabolome. Results showed that a total of 254 flavonoid metabolites (including 147 flavone, 39 flavonol, 21 flavanone, 24 anthocyanins, 8 isoflavone, and 15 polyphenol) were identified. The total flavonoid content of peels from DHP was the highest. DHP could be clearly separated from other samples through clustering analysis and principal component analysis (PCA). Further, 169 different flavonoid metabolites were observed between DHP peels and the other four citrus peels, and 26 down-regulated differential metabolites displayed important biological activities in DHP. At the same time, a unique flavonoid component, tricin 4′-*O*-syringyl alcohol, was only found in DHP, which could be used as a marker to distinguish between other varieties. This work might facilitate a better understanding of flavonoid metabolites between DHP peels and the other four citrus peels and provide a reference for its sufficient utilization in the future.

## 1. Introduction

Citrus reticulata “Dahongpao” (DHP) is an ancient citrus variety. Its fruits and peels are rich in nutrients and bioactive compounds, and are very popular because of their antioxidant properties and extremely high medicinal value [1]. In recent years, due to the development of new citrus varieties, DHP has been almost completely replaced by various hybrid mandarins in China, however the health promotion or medicinal values of the “Dahongpao” tangerine have attracted more and more attention from researchers [2]. It was reported that the total flavonoid content of DHP peel was one of the highest in citrus varieties [3], and it has been included in the key protection of citrus varieties in China. Previous studies have shown that flavonoids are involved in many biological functions and have important health-related roles, such as anti-oxidative [4], anti-diabetic [5], anti-inflammatory [6], and anti-hypertensive [7] activities. Therefore, systematic analysis of the flavonoids and screening the unique components in DHP peels are of great significance for the sufficient utilization and development of DHP.

Traditionally, high-performance liquid chromatography (HPLC) coupled with DAD, HPLC coupled with MS, or even UPLC-Q-TOF-MS, were often adopted to analyze the components of citrus varieties. Green et al. [8] studied six polymethoxylated flavones in peels of selected Jamaican and Mexican citrus cultivars by HPLC; Zhang et al. [9] identified 28 flavonoid compositions of flavedo and juice using an HPLC-DAD-MS/MS system; Zhao et al. [10] used UPLC-Q-TOF-MS to analyze 81 phytochemical components in the peel of Chinese wild Citrus Mangshanju. Zhao et al. detected 92 compounds in the peels of six oranges and 10 mandarins using UHPLC-Q-TOF-MS; Li et al. [2] established a preparative separation method, which could simultaneously isolate the polymethoxylated flavones (PMFs) from the peel of DHP using macroporous adsorptive resins (MARs) combined with prep-HPLC. 

As a whole, the analysis and detection of bioactive compounds in citrus varieties has been studied by many researchers. However, all these methods have the shortcoming of only checking a small number of compounds, which cannot profile the chemical compositions of citrus species in a systematic and comprehensive way. Less than 100 compounds in citrus peel have been reported in the existing literature. In recent years, UPLC-ESI-MS/MS-based, widely targeted metabolome has become very popular in the field of analysis and identification of plant metabolites due to the advantages of high throughput, fast separation, high sensitivity, and wide coverage. At present, this method has been widely applied in plant metabolites analysis in maize [11], rice [12,13,14], tomato [15,16], potato [17], and other plants [18,19]. So far, it is an effective method to thoroughly understand plant secondary metabolites.

Researchers have reported that the types of bioactive compounds in citrus peels are often higher than that in pulp and seeds [20]. So, those citrus peels can be totally regarded as a kind of nutraceutical [21] or functional food to complement and help with the treatment of pathologies [22,23]. In this study, five important citrus varieties, including two mandarins (DHP and BZH), one pummelo (HZY), and two oranges (TC and YCC), were selected as experimental materials in order to have a better understanding of the flavonoids profiling of DHP peel and to clarify the unique flavonoids. A UPLC-ESI-MS/MS method was used for profiling the flavonoid metabolites in the peel of DHP and the other four citrus varieties. These results might facilitate a better understanding of flavonoid metabolites between DHP peels and four other citrus peels and provide a reference for their sufficient utilization in the future.

## 2. Results

### 2.1. Determination of Total Flavonoid Content

In this study, the total flavonoid content was determined from the *Citrus reticulata* “Dahongpao” (DHP) sample and four other citrus varieties, including *Citrus grandis Tomentosa* (HZY), *Citrus ichangensis Swingle* (YCC), *Citrus sinensis (L.) Osbeck* (TC), and *Citrus reticulata* “Buzhihuo” (BZH). The method of ordinary one-way ANOVA was used to analyse the data. The flavonoid content of the peels from DHP was significantly higher (*p* < 0.05) than that of HZY, YCC, TC, and BZH, reaching 122.5 mg/g (Figure 1). The flavonoid content of BZH was the second highest at 50.1 mg/g. The experimental results were consistent with the literature reported [4].

### 2.2. Metabolic Profiling 

The flavonoid metabolites of the citrus peels from DHP and four other citrus varieties were investigated based on UPLC-ESI-MS/MS and databases. In the present study, 254 flavonoid metabolites were identified (Appendix A), including 147 flavone, 39 flavonol, 21 flavanone, 24 anthocyanins, 8 isoflavone, and 15 polyphenol. In the heatmap (Appendix A), the content of flavonoid metabolites in DHP contrasted with YCC and HZY varied greatly, whereas the contents of flavonoid metabolites among YCC and HZY were basically consistent. Although DHP, BZH, and TC were grouped into one category, the content of metabolites was also quite different from each other on the heatmap. This finding was demonstrated by clustering analysis of the samples and showed that DHP was clearly distinguished from the other four citrus varieties. 

### 2.3. Differential Flavonoid Metabolite Analysis Based on PCA

Principle component analysis (PCA) uses several principal components to reveal the overall metabolic differences among the groups and the variability between the intra-group samples. In this study, two principal components, PC1 and PC2, were extracted and were 40.04% and 24.88%, respectively; moreover, the cumulative contribution rate reached 64.92%. In the PCA score plot, DHP, YCC, TC, HZY, and BZH were clearly separated, and the repeated samples were compactly gathered together (Figure 2), thus indicating that the experiment was reproducible and reliable. In the PCA 3D map, the clustering of samples could be seen more intuitively. Through PCA analysis, it was found that the difference of flavonoid metabolic components among samples might be the difference between DHP and other citrus varieties. 

### 2.4. Differential Flavonoid Metabolite Analysis Based on OPLS-DA

Orthogonal signal correction and partial least squares-discriminant analysis (OPLS-DA) is an effective method for screening differential metabolites because it can maximize the difference between groups. Q^2^ is an important parameter for evaluating the models in OPLS-DA. Q^2^ values greater than 0.9 indicate an excellent model. In this study, the OPLS-DA model compared the flavonoid metabolite content of the samples in pairs to evaluate the difference between DHP and YCC (R^2^X = 0.964, R^2^Y = 1, Q^2^ = 1), between DHP and TC (R^2^X = 0.948, R^2^Y = 1, Q^2^ = 0.999), between DHP and HZY (R^2^X = 0.964, R^2^Y = 1, Q^2^ = 1), and between DHP and BZH (R^2^X = 0.864, R^2^Y = 1, Q^2^ = 0.998). The Q^2^ values of all comparison groups exceeded 0.9 (Figure 3), thus demonstrating that these models were stable and reliable and could be used to further screen for differential flavonoid metabolites.

### 2.5. Differential Flavonoid Metabolite Screening, Functional Annotation, and Enrichment Analysis

Differential flavonoid metabolites were screened for each comparison group by combining the fold change and variable importance in project (VIP) values of the OPLS-DA model. The criteria for screening included a fold change value of ≥2 or ≤0.5 and a VIP value of ≥1. The screening results are shown in Appendix A. A volcanic plot provides a quick way to see the differences in metabolite expression levels in two samples, as well as the statistical significance of the differences (Figure 4). There were 52 significantly different flavonoid metabolites between DHP and BZH (21 down-regulated, 31 up-regulated), 100 between DHP and HZY (64 down-regulated, 36 up-regulated), 68 between DHP and TC (41 down-regulated, 27 up-regulated), and 97 between DHP and YCC (65 down-regulated, 32 up-regulated). Compared with DHP, the flavonoid metabolites of BZH were up-regulated, and HZY, TC, and YCC were down-regulated. Further, 169 differential metabolites were observed in all the comparison groups and 26 down-regulated differential metabolites display important biological activities in DHP (Table 1). After taking an intersection of each comparison group in a Venn diagram (Figure 5a), 8 common differential metabolites were observed, and each comparison group had its unique differential metabolites. Therefore, differential metabolites could clearly distinguish DHP from other citrus varieties.

The differential flavonoid metabolites from each comparison group were annotated by the Kyoto Encyclopedia of Genes and Genomes (KEGG) database (Appendix A). The KEGG classification results and enrichment analysis (Figure 5c–f) indicate that the differential flavonoid metabolites of the comparison groups were involved in isoflavonoid biosynthesis, flavonoid biosynthesis, flavone and flavonol biosynthesis, and anthocyanin biosynthesis. 

## 3. Discussion

The analysis of the experimental data showed that the peels of DHP were rich in flavonoids, and the flavonoid content could reach 122.5 mg/g, which is significantly higher than that of HZY, YCC, TC, and BZH, however the utilization is extremely inadequate. Thus, this study aimed to provide a theoretical basis for the utilization of DHP peels. The enrichment analysis of the differential metabolites revealed that differential metabolites were mainly involved in isoflavonoid biosynthesis, flavonoid biosynthesis, flavone and flavonol biosynthesis, and anthocyanin biosynthesis.

Isoflavones are generally exclusively present in legumes, such as soybeans, and play important roles in plant defense and nodules [24]. In the present study, eight isoflavones were detected in orange peels, of which only the compound prunetin and biochanin A were down-regulated, whereas other compounds were up-regulated. Prunetin is always found in herbs and spices, like Prunus species or Glycyrrhiza glabra [25]. Biochanin A is an isoflavone derivative isolated from red clover *Trifolium pratense* with anticarcinogenic properties; it has estrogen-like effects, can inhibit the rise of cholesterol, however it also has anti-fungal and anti-tumor effects [26]. Five flavonoid compounds were detected in flavonoid biosynthesis in all the samples. The most interesting thing was that all of them were down-regulated when other samples were compared with DHP. Dihydromyricetin has many peculiar effects, such as scavenging free radicals, antioxidant, anti-thrombosis, anti-cancer, and anti-inflammatory actions [27,28,29]. Dihydroquercetin, as an important flavonoid, exists in many plants and has a high content of *larch*, especially *Pinus elliottii* [30]. In recent years, dihydroquercetin has also been found in fruits, especially grapes [31] and oranges [32]. Anthocyanin is a kind of water-soluble natural pigment widely found in plants in nature, which is also a powerful antioxidant that can protect against harmful substances [33]. Compared with DHP, most of the anthocyanin metabolites in the peels of other samples were down-regulated. Cyanidin 3-*O*-galactoside was the only differential anthocyanin detected in the comparison group DHP versus BZH. Comparison groups showed that 117 overlap metabolites and 11 unique metabolites were detected (Figure 5b). They were Tricin 4′-*O*-syringyl alcohol, Tricin *O*-rhamnosyl-*O*-malonylhexoside, Chrysoeriol *O*-hexosyl-*O*-pentoside, 8-*C*-hexosyl-chrysoeriol *O*-feruloylhexoside, Apigenin *O*-hexosyl-*O*-pentoside, Apigenin *O*-hexosyl-*O*-rutinoside, Chrysoeriol 5-*O*-hexoside, Daidzein 7-*O*-glucoside, Gossypol, Pedalitin, and Demethoxycurcumin. At the same time, Tricin 4′-*O*-syringyl alcohol was only detected in DHP, and the compounds Daidzein 7-*O*-glucoside and Demethoxycurcumin were reported to have pharmacological activities in the literature [34,35].

The identification of closely-related citrus varieties is controversial because of substantial interspecific hybridization, which has resulted in several clonally propagated and cultivated accessions [36]. Some hybrids mandarins are closely-related and have very small differences, so it is very difficult to identify them. In this paper, DHP and BZH are two closely-related citrus varieties. Through clustering analysis, principal component analysis (PCA), and orthogonal signal correction and partial least squares-discriminant analysis (OPLS-DA), DHP and BZH were clearly separated. Therefore, metabolites analysis based on the UPLC-ESI-MS/MS platform may be an effective method to distinguish closely-related species.

## 4. Materials and Methods

### 4.1. Plant Materials 

Five citrus varieties were collected from the Citrus Research Institute of the Chinese Academy of Agricultural Sciences (106°24′ E, 29°59′ N, altitude 175 m). They were *Citrus reticulata* “Dahongpao” (DHP), *Citrus grandis Tomentosa* (HZY), *Citrus ichangensis Swingle* (YCC), *Citrus sinensis (L.) Osbeck* (TC), and *Citrus reticulata* “Buzhihuo” (BZH), respectively. The citrus peels were frozen in liquid nitrogen immediately after collection and stored at −80 °C. 

### 4.2. Sample Preparation and Extraction 

The freeze-dried citrus peels were crushed using a mixer mill (MM400, Retsch, Laichi, Germany) with a zirconia bead for 1.5 min at 30 Hz. Then, 100 mg of powder was weighed and extracted overnight at 4 °C with 1.0 mL 70% aqueous methanol. Following centrifugation at 10,000× *g* for 10 min, the extracts were absorbed (CNWBOND Carbon-GCB SPE Cartridge, 250 mg, 3 mL; ANPEL, Shanghai, China) and filtrated (SCAA-104, 0.22 μm pore size; ANPEL, Shanghai, China) before LC-MS analysis.

### 4.3. UPLC Conditions 

The sample extracts were analyzed using an LC-ESI-MS/MS system (UPLC, Shim-pack UFLC SHIMADZU CBM30A system; MS, Applied Biosystems 6500 QTRAP). The UPLC conditions were performed according to the method by Wang et al. [37]. The analytical conditions were as follows, UPLC: column, Waters ACQUITY UPLC HSS T3 C18 (1.8 µm, 2.1 mm × 100 mm, Milford, MA, USA); solvent system, water (0.04% acetic acid): acetonitrile (0.04% acetic acid); gradient program, 100:0 *v*/*v* at 0 min, 5:95 *v*/*v* at 11.0 min, 5:95 *v*/*v* at 12.0 min, 95:5 *v*/*v* at 12.1 min, 95:5 *v*/*v* at 15.0 min; flow rate, 0.40 mL/min; temperature, 40 °C; injection volume: 2 μL. The effluent was alternatively connected to an ESI-triple quadrupole-linear ion trap (Q TRAP)-MS.

### 4.4. ESI-Q TRAP-MS/MS

Mass spectrometry followed the method of Chen et al. [13]. LIT and triple quadrupole (QQQ) scans were acquired on a triple quadrupole-linear ion trap mass spectrometer (Q TRAP), API 6500 Q TRAP LC/MS/MS System, equipped with an ESI Turbo Ion-Spray interface, operating in a positive ion mode and controlled by Analyst 1.6.3 software (AB Sciex, Waltham, MA, USA). The ESI source operation parameters were as follows: ion source, turbo spray; source temperature 500 °C; ion spray voltage (IS) 5500 V; ion source gas I (GSI), gas II (GSII), and curtain gas (CUR) were set at 55, 60, and 25.0 psi, respectively; the collision gas (CAD) was high. Instrument tuning and mass calibration were performed with 10 and 100 μmol/L polypropylene glycol solutions in QQQ and LIT modes, respectively. QQQ scans were acquired as MRM experiments with collision gas (nitrogen) set to 5 psi. DP and CE for individual MRM transitions was done with further DP and CE optimization. A specific set of MRM transitions were monitored for each period according to the metabolites eluted within this period.

### 4.5. Determination of Total Flavonoids Content 

Approximately 2.0 g of citrus peels from each sample were freeze-dried by using a vacuum freeze dryer and then ground into powder using a mortar. The total flavonoid content of 0.02 g citrus peels was determined in accordance with the protocol of Plant Flavonoids Test kit (Suzhou Comin Biotechnology Co., Ltd., Suzhou, China).

### 4.6. Qualitative and Quantitative Analysis of Metabolites

Qualitative and quantitative analysis of metabolites were performed according to the method by Wang et al. [37]. Based on the self-built database MWDB (Metware Biotechnology Co., Ltd. Wuhan, China) and public database of metabolite information, the metabolites of the samples were qualitatively and quantitatively analyzed by mass spectrometry. The characteristic ions of each substance were screened out by the triple quadrupole rod, and the signal strength of the characteristic ions were obtained in the detector. The mass spectrometry file under the sample was opened with MultiaQuant software 3.0.3 to carry out the integration and correction of chromatographic peaks, and the relative content of the corresponding substances in the peak area of each chromatographic peak were calculated. Finally, all chromatographic peak area integral data were derived. In order to compare the contents of each metabolite in different samples, we calibrated the mass spectrum peaks detected by each metabolite in different samples based on the information of metabolite retention time and peak pattern. Thus, the accuracy of the qualitative and quantitative analysis was further ensured. 

### 4.7. Sample Quality Control Analysis 

The high stability of the instrument provided an important guarantee for the repeatability and reliability of the data. In order to test the stability of the instrument, one quality control sample was inserted into every 10 samples in the process of instrument analysis to monitor the repeatability of the analysis process, and the repeatability of metabolite extraction and detection was judged by overlapping the analysis of total ion flow maps of different quality control samples.

### 4.8. Statistical Analysis 

Three biological replicates were performed in each experiment. Cluster analysis, PCA, and OPLS-DA were carried out by using R (http://www.r-project.org/) in accordance with previously described methods [38].

## 5. Conclusion

In this paper, the flavonoid metabolites in the peels of DHP and the other four citrus varieties were systematically analyzed and identified using a UPLC-ESI-MS/MS-based, widely targeted metabolome. A total of 252 flavonoid metabolites were detected, 117 of which were compounds shared by all samples, and 169 differential metabolites were observed in all the comparison groups, 8 of which were common differential metabolites. At the same time, a total of 11 unique metabolites were detected in all the samples, of which Tricin 4′-*O*-syringyl alcohol only existed in DHP and could be used as a marker to distinguish from the other four citrus varieties. Furthermore, the total flavonoids content of the peels from DHP was higher than all the other samples and 26 down-regulated differential metabolites displayed important biological activities in DHP. Hence, the peels from DHP have considerable potential for development and utilization in the future, such as citrus peel tea or health food.

## Figures and Tables

**Figure 1 molecules-24-02680-f001:**
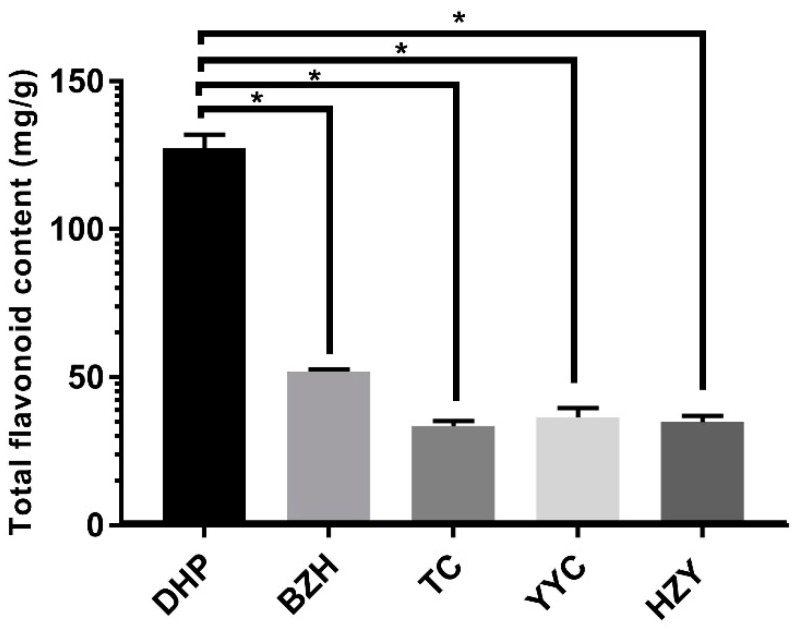
Total flavonoid contents of all the samples in the experiment. The five-pointed star above the histogram indicates the statistical significance at the level of 0.05 (*p* < 0.05).

**Figure 2 molecules-24-02680-f002:**
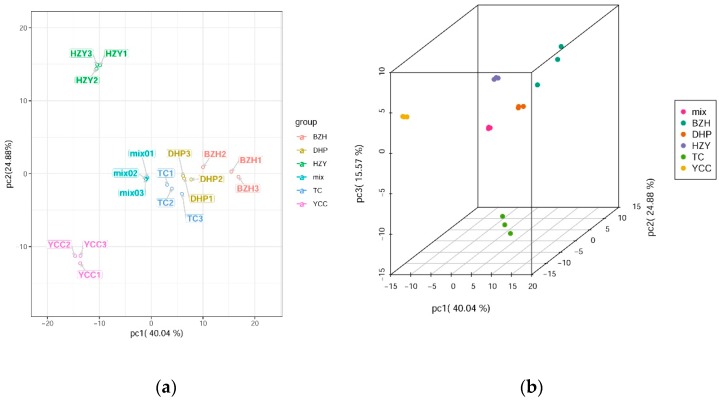
Differential flavonoid metabolite analysis on the basis of principal component (PCA). (**a**) PCA score plot; (**b**) PCA 3D plot.

**Figure 3 molecules-24-02680-f003:**
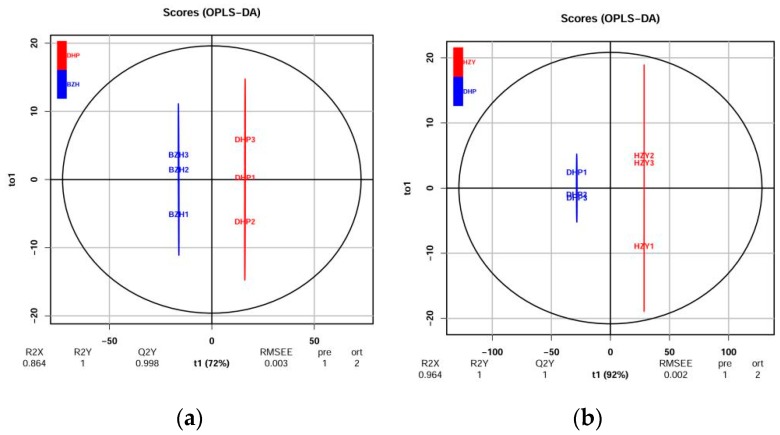
Differential flavonoid metabolite analysis on the basis of orthogonal signal correction and partial least squares-discriminant analysis (OPLS-DA). (**a**–**d**) OPLS-DA model plots for the comparison group DHP versus BZH, DHP versus HZY, DHP versus TC, DHP versus YCC, respectively.

**Figure 4 molecules-24-02680-f004:**
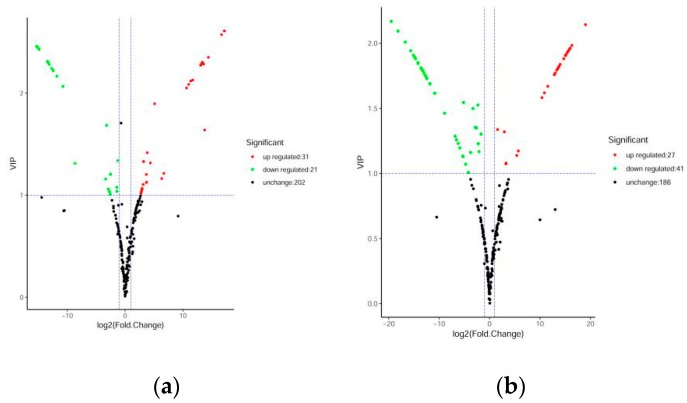
Volcanic plots of differential metabolites. (**a**) DHP versus BZH; (**b**) DHP versus TC; (**c**) DHP versus HZY; (**d**) DHP versus YCC. Each point in the volcanic plot represents a metabolite, the abscissa represents the logarithm of the quantitative difference multiples of a metabolite in two samples, and the ordinate represents the variable importance in project (VIP) value. The larger the abscissa absolute value is, the more significant the differential expression is, and the more reliable the screened differential expression metabolites are. The green dots in the figure represent down-regulated differentially expressed metabolites, the red dots represent up-regulated differentially expressed metabolites, and the black dots represent metabolites detected but that are not significantly different.

**Figure 5 molecules-24-02680-f005:**
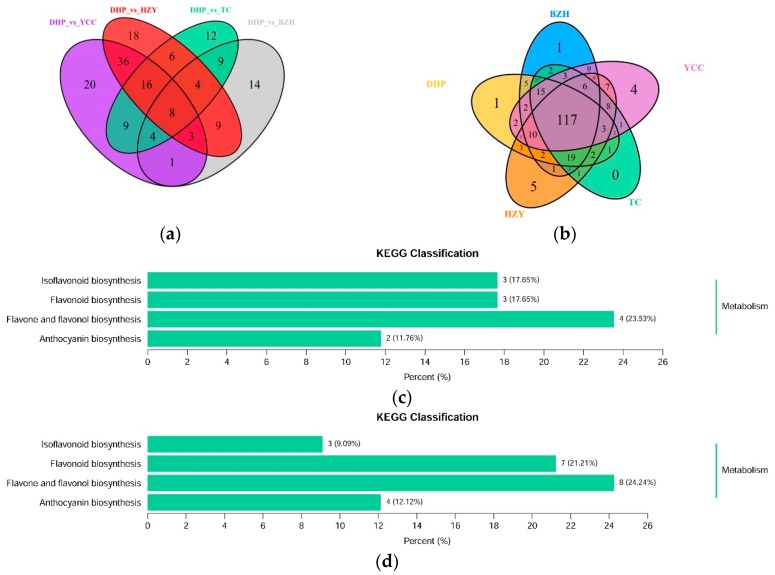
Venn diagram of differential flavonoid metabolites for each comparison group and Kyoto Encyclopedia of Genes and Genomes (KEGG) classification results. (**a**) Venn diagram shows the overlapping and unique differential metabolites amongst the comparison groups; (**b**) Venn diagram shows the overlapping and unique metabolites amongst the comparison groups; (**c**)–(**f**) The differential metabolites KEGG classification of the comparison group DHP versus BZH, DHP versus HZY, DHP versus TC, and DHP versus YCC, respectively.

**Table 1 molecules-24-02680-t001:** A list of biologically active down-regulated flavonoid metabolites detected in “Dahongpao” (DHP).

All_ID	Q1 (Da)	Q3 (Da)	Rt (min)	Molecular Weight (Da)	Compounds
pmb2850	329	314	5.73	330	Tricin
pme0372	435.1	273	4.22	434.12	Naringenin 7-*O*-glucoside (Prunin)
pme0379	271.1	153	5.63	270.05	Apigenin
pme1399	355.2	178.5	8.4	354.15	Xanthohumol
pme1562	441.3	169	3.89	442.3	Epicatechin gallate (ECG)
pme2293	367.1	149	7.38	368.13	Curcumin
pme2898	321.1	303	3.52	320.05	Dihydromyricetin
pme2949	609.2	301	4.08	610.19	Hesperetin 7-rutinoside (Hesperidin)
pmf0234	563.1	443.1	3.53	564.15	Isoschaftoside
pmf0360	449.1	151	3.97	450.12	Astilbin
pmf0381	563.1	353	3.5	564.15	Vicenin-3
pmf0417	595.2	287	3.7	596.17	Eriocitrin
pmf0458	295.2	277	7.08	294.18	6-Gingerol
pmf0567	299.1	211	5.72	300.06	Tectorigenin
pmf0584	437.1	275	4.37	436.14	Phloridzin
pme0330	579.2	271	4.17	580.18	Naringenin 7-*O*-neohesperidoside (Naringin)
pme0369	593.2	285	3.83	594.16	Kaempferol 3-*O*-rutinoside (Nicotiflorin)
pme1541	283.1	268	7.06	284.07	Acacetin
pme2977	347.3	285	6.66	346.25	Troxerutin (Trihydroxyethyl rutin)
pme3212	465	303	3.87	464.1	Quercetin 3-*O*-glucoside (Isotrifoliin)
pme3250	285	270	7	284.07	Biochanin A
pme3292	283	268	6.97	284.07	Prunetin
pmf0005	579.2	271	4.05	580.53	Narirutin
pmf0208	465.1	303	3.9	464.1	Isoquercitroside
pmf0236	563.1	241.2	4.68	564.13	Theaflavin
pmf0549	607.2	299	4.14	608.17	Diosmin

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
