# Peer review of "Analysis of Flavonoid Metabolites in Citrus Peels (*Citrus reticulata* “Dahongpao”) Using UPLC-ESI-MS/MS"

_molecules, 2019, doi:10.3390/molecules24152680_

Round 1
Reviewer 1 Report
Abstract section - The authors claim that considerable 21 different flavonoids metabolites were observed between DHP peels and other four citrus peels, some of them displayed with important biological activities. Please clearly indicate which are important metabolites.
Abstract section - The authors claim that they found the 23 unique flavonoid component of DHP, which could be used as a marker to distinguish with 24 other varieties. Please point out those compounds directly.
The authors claim that results shown 17 that a total of 254 flavonoid metabolites (including 147 flavone, 39 flavonol, 21 flavanone, 18 24 anthocyanins, 8 isoflavone, and 15 polyphenol) were identified. Please provide an iconic mass spectra for proving it.
Conclusion section - New findings are limited, and most of which seem to be known knowledge, e.g. citrus peels were rich in flavonoids, flavonoid etabolites varied widely in different samples, and had their own unique metabolites, the peels from DHP have considerable potential for development and utilization in the future, such as citrus peel tea or health food (see Introduction section: Citrus reticulata 'Dahongpao' (DHP) is an ancient citrus variety, its fruits and peels are rich in nutrients and bioactive compounds, and are very popular because of their antioxidant properties and extremely high medicinal value.......Also see Reference 1: Li, Z.; Zhao, Z.; Zhou, Z. Simultaneous Separation and Purification of Five Polymethoxylated Flavones from "Dahongpao" Tangerine (Citrus tangerina Tanaka) Using Macroporous Adsorptive Resins Combined with Prep-HPLC. Molecules. 2018, 23, 2660.).
Please organize and describe a main proposal or suggestion for the most important finding in this work.
Reviewer 2 Report
The paper is interesting. However, it needs a lot of improvememt befote it can be accepted for publication.
Language must be revised by a native English speaker (This must be certified).
Figures 2, 3, 4 and 5 are very bad. They need a lot of work. Letters and numbers can not be seem. All figures must be re-edited.
Please check the format. See Section "2 results"? Please take your time to "really" verify the paper is in good shape.
The authors must explain the differences between this report and previous ones on flavonoids for these species (there are many).
Avoid using first person on writting (we, I).
Statistical analysis did not mention anything on how to compare means. It should also state the significance level.
Introduction and Discussion are poor. It can be improved. Provide a table on main components that have been reported elsewhere.
Reviewer 3 Report
The manuscript entitled: "Analysis of flavonoids metabolites in citrus peels (Citrus reticulata "Dahongpao") using UPLC-ESI-MS/MS" reports data on characterization with reference/comparison to five (see Abstract) Citrus varieties not only one as per the manuscript title. For this reason title of the manuscript should be adjusted/modified accordingly. The results are properly described, and the content is appealing, due also to the interest raising on flavonoids as antioxidants for medicinal and nutraceutical use. This last aspect, the nutraceutical use, which is growing in the interest worldwide from scientific and industrial point of view, should be also evidenced in the Introduction section of the manuscript. Appropriate references should be cited.
In the following, the references needed to assess the overall context are indicated:
Santini et al. Nutraceuticals: shedding light on the grey area between pharmaceuticals and food. Expert Review of Clinical Pharmacology, 2018, 11(6), 545 547.
Daliu et al. From pharmaceuticals to nutraceuticals: bridging disease prevention and management. Expert Review of Clinical Pharmacology, 2019. 12(1), 1-7.
Santini et al. State of the art of Ready-to-Use Therapeutic Food: a tool for nutraceuticals addition to foodstuff. Food Chem., 2013. 15, 140(4):843-849.
The experimental part is properly assessed and described. The results are properly justified. A Conclusion section with the summary of the results observed and the perspective point of view on the topic studied by the Authors would be needed for the proper completeness of the manuscript. Minor English check would be advisable.
Round 2
Reviewer 1 Report
The manuscript has been improved according to the comments. It can be accepted for publication.
Reviewer 2 Report
The authors made appropriate changes in the revised version. It should be published.